# Population Pharmacogenomics for Health Equity

**DOI:** 10.3390/genes14101840

**Published:** 2023-09-22

**Authors:** I. King Jordan, Shivam Sharma, Leonardo Mariño-Ramírez

**Affiliations:** 1School of Biological Sciences, Georgia Institute of Technology, Atlanta, GA 30332, USA; shivamsharma13@gatech.edu; 2National Institute on Minority Health and Health Disparities, National Institutes of Health, Bethesda, MD 20892, USA; marino@nih.gov

**Keywords:** pharmacogenomics, health equity, health disparities, race, ethnicity, ancestry, population, adverse drug reactions

## Abstract

Health equity means the opportunity for all people and populations to attain optimal health, and it requires intentional efforts to promote fairness in patient treatments and outcomes. Pharmacogenomic variants are genetic differences associated with how patients respond to medications, and their presence can inform treatment decisions. In this perspective, we contend that the study of pharmacogenomic variation within and between human populations—population pharmacogenomics—can and should be leveraged in support of health equity. The key observation in support of this contention is that racial and ethnic groups exhibit pronounced differences in the frequencies of numerous pharmacogenomic variants, with direct implications for clinical practice. The use of race and ethnicity to stratify pharmacogenomic risk provides a means to avoid potential harm caused by biases introduced when treatment regimens do not consider genetic differences between population groups, particularly when majority group genetic profiles are assumed to hold for minority groups. We focus on the mitigation of adverse drug reactions as an area where population pharmacogenomics can have a direct and immediate impact on public health.

## 1. Introduction

### Population Pharmacogenomics Can and Should Be Leveraged for Health Equity—Improved Health Outcomes for All People Everywhere

The United States (US) National Institute on Minority Health and Health Disparities (NIMHD) defines health equity as the assurance that “all individuals or populations have optimal opportunities to attain the best health possible” [1]. The converse of health equity is health disparities, which are defined by NIMHD as differences in health outcomes that affect disadvantaged populations, and the elimination of health disparities is a key component of health equity [2]. It should be stressed, however, that the health equity paradigm goes beyond just eliminating health disparities to include more aspirational goals of achieving optimal health for all populations.

NIMHD states that the application of a health equity lens to biomedical research requires intentional efforts to “promote fairness, opportunity, quality, and social justice in treatments and outcomes” [1]. This perspective article is focused on how population pharmacogenomics research can be deployed as one such intentional effort in support of health equity. We define population pharmacogenomics as the study of pharmacogenetic variation within and between populations, where pharmacogenetic variation refers to genetic differences associated with the response to medications, and populations are delineated as groups of people that share common characteristics, such as race, ethnicity, or ancestry. We explore how population pharmacogenomics research can and should be leveraged to facilitate health equity through a focus on medical treatments that consider the genetic variation of socially disadvantaged racial and ethnic groups that experience health disparities. We highlight examples from our own research on the relationship between race, ethnicity, ancestry, and pharmacogenomic variation in cosmopolitan countries of the global north, the US and the United Kingdom (UK), and in the South American country of Colombia, to illuminate the potential of pharmacogenomics to promote health equity.

## 2. Race, Ethnicity, and Pharmacogenomic Variation

### Socially Defined Racial and Ethnic Groups Exhibit Differences in the Frequencies of Pharmacogenomic Variants, with Direct Implications for Health Equity

The US National Academies of Science Engineering and Medicine (NASEM) recently released a comprehensive and detailed report on the use of race, ethnicity, and ancestry as population descriptors in genomics research [3]. The overarching recommendation of this report was that researchers should carefully consider and justify how and why they use race and ethnicity as population labels in their work, a recommendation with which we fully agree. The report emphasizes that race and ethnicity are social concepts, which are nevertheless often used as surrogates for describing human genetic differences. The use of race and ethnicity in this way can be problematic in the sense that it may reinforce the erroneous notion of discrete human groups with innate differences, while failing to capture complex patterns of global genetic variation.

Our use of race and ethnicity population descriptors in this perspective, and the broader relevance of these group labels to pharmacogenomics, is rooted in both empirical and pragmatic considerations. With respect to the empirical justification for the use of race and ethnicity descriptors in pharmacogenomic research, we recognize that racial and ethnic groups are socially constructed. By this, we mean that the definitions of these groups, the boundaries around the groups which in turn define their composition, are determined by human actors. In the case of the US, the UK, and Colombia, government bureaucrats decide on and enumerate group labels and their definitions [4,5,6,7]. Racial and ethnic groups differ across space and time, depending on the demographic composition of the societies in which they are used. The US uses a system that combines racial groups, defined by shared ancestral origins, and ethnic groups defined by shared culture. The UK defines ethnic groups based on individuals’ national origins, and Colombia uses the term *etnia* (Spanish for ethnicity) for ethnic minority groups and *sin pertenencia étnica* (Spanish for no ethnicity) for the majority population, which shows European and American admixture. In the US, the definitions of racial and ethnic groups have changed twenty times since they were first used in the late 18th-century census, and similar changes have occurred over time in the UK and Colombia [8].

Nevertheless, the social and genetic dimensions of race and ethnicity are not mutually exclusive, and they can both be important for health outcomes. In particular, there is abundant evidence that socially defined racial and ethnic groups can show pronounced differences in the allele frequencies of genetic variants, including groups of variants that collectively influence polygenic traits, many of which may have clinical relevance [9,10,11,12,13,14]. It should be emphasized that these are differences of quantity rather than differences of kind; alleles differ in relative frequencies between groups with almost no alleles found universally present in one group and universally absent in another group [15,16,17]. Allele frequency differences between groups are a consequence of the fact that individuals’ racial and ethnic identities are tied to their ancestral origins, which are in turn correlated with genetic differences. This fact is underscored by the NASEM report’s definition of race, ethnicity, and ancestry as “descent-associated” groups composed of members who share characteristics based on common origins [3]. Groups with ancestors that evolved in different parts of the world were once reproductively isolated, and they may remain so owing to cultural differences, and thus accumulated genetic differences over time [18,19,20]. The resulting differences in the allele frequencies of pharmacogenomic variants among racial and ethnic groups have direct relevance to medical treatment decisions.

As a more pragmatic consideration, race and ethnicity are readily available for use by clinicians, whereas patient genetic data is substantially harder to come by. Later in this perspective, we discuss how direct genetic assays capture pharmacogenomic variation far more reliably than race and ethnicity, which are poor proxies for individual-level genetic variants, but patients from socially disadvantaged groups are less likely to have access to these kinds of data than patients from majority groups. Finally, the use of race and ethnicity in pharmacogenomic research and clinical applications can help to avoid the kind of bias that arises when drug development and treatment decisions are made based on research conducted primarily on members of the majority group. If there are meaningful differences in the frequencies of pharmacogenomic variants between race and ethnicity groups, treatment strategies tailored to the majority group have the potential to do real harm to members of minority racial and ethnic groups. We believe that consideration of race and ethnicity when making treatment decisions, as we elaborate on below, can help to avoid these kinds of biases and their downstream consequences.

Here, we briefly review examples of pharmacogenomic differences among racial and ethnic groups from our own work in the US, the UK, and Colombia [21,22,23,24,25]. Most recently, we used population biobanks to interrogate the relationship between race, ethnicity, and pharmacogenomic variation in the US and the UK [25]. Socially defined race and ethnicity were found to be highly correlated with genome-wide patterns of pharmacogenomic variation, in support of the relevance of these categories for patient stratification. The cohorts for this study were taken from the US All of Us Research Program (*All of Us*; n = 65,120; Figure 1A) and the UK Biobank (*UKB*; n = 31,396; Figure 1B). Machine learning classifiers were used to predict biobank participants’ self-identified race and ethnicity (SIRE) based on patterns of variation for 6311 pharmacogenomic variants in *All of Us* (Figure 1C) and 5966 pharmacogenomic variants in *UKB* (Figure 1D). Pharmacogenomic variation was found to predict participants’ SIRE with 92.1% accuracy for *All of Us* and 94.3% accuracy for *UKB*. The highest group-specific prediction accuracy was 99.0% for the White group in *UKB*, and the lowest group-specific prediction accuracy was 92.9% for the Hispanic group in *All of Us*. We also found numerous individual pharmacogenomic variants with large allele frequency differences between racial and ethnic groups in these cohorts, consistent with previous studies.

In 2020, we analyzed a cohort of 8628 participants from the US Health and Retirement Study (*HRS*) to compare the ability of SIRE versus genetic ancestry (GA) to stratify pharmacogenomic variants [22]. We hypothesized that GA would provide greater resolution for pharmacogenomic variant stratification compared to SIRE, given that SIRE is socially defined whereas GA is a characteristic of the genome, which can be characterized objectively and with precision. Nevertheless, GA was found to predict SIRE with >96% accuracy, and thus GA provided a negligible increase in resolution for pharmacogenomic variant stratification. This study also confirmed the long-held notion that the majority of human genetic variation is found within rather than between racial and ethnic groups, a fact that is often taken to argue against the relevance of race and ethnicity to human genetic variation. Despite this pattern of pharmacogenomic variance partitioning, numerous pharmacogenomic variants were found to show significant frequency differences among SIRE groups, and genome-wide patterns of pharmacogenomic variation were highly concordant with SIRE. This study underscored the clinical relevance of SIRE for the stratification of pharmacogenomic risk among groups, while also showing far less utility of SIRE as a proxy for individual-level pharmacogenomic stratification. We explore the distinction between individual-level versus population-level pharmacogenomic risk stratification further in the following section on precision medicine versus precision public health.

In 2019, data from the ChocoGen research project was used to investigate population pharmacogenomics in the Colombian populations of Antioquia and Chocó in support of health equity in Colombia [21]. Antioquia and Chocó are neighboring departments (states) with distinct ancestry profiles: European followed by American ancestry in Antioquia compared to primarily African ancestry in Chocó. We found numerous pharmacogenomic variants with highly divergent allele frequencies between these two Colombian populations, with differences that were correlated to their distinct ancestral backgrounds. We used these findings, working with local partners in Colombia, to develop and validate allele-specific PCR assays to test patients for population-specific pharmacogenomic variants. This approach allowed our Colombian partners to focus local resources on the populations where they are more likely to yield a return on investment, thereby serving as an example of how population pharmacogenomics can be leveraged to support precision medicine in resource-limited settings.

## 3. Global and Local Views on Race, Ethnicity, and Genetics

### Race and Ethnicity Stratify Pharmacogenomic Variation at the Local Level but Do Not Represent Natural Groups That Correspond to Global Patterns of Human Genetic Variation

The racial and ethnic stratification of pharmacogenomic variation described in the previous section does not negate the fact that these groups are social constructs, which do not map well to global patterns of human genetic variation. At the global level, human genetic variation is largely distributed as a continuum, based on reproductive isolation by distance, with discontinuities introduced by major geographic barriers, like oceans, mountain ranges, and deserts [26]. Furthermore, the vast majority of human genetic variation can be found within the African continent, consistent with the fact that modern humans (Homo sapiens) evolved in Africa for more than 200,000 years before migrating out of Africa and populating the rest of the world [27,28,29]. The deepest human lineages divide the Khoisan groups from all other human populations, with a divergence time of ~200,000 years, followed by the Rainforest Hunter-Gather split at ~150,000 years ago. West Africans and Eurasians who migrated out of Africa share a substantially more recent divergence ~75,000 years ago (note that all dates are approximate and subject to refinement). This means that people of the African diaspora in the US, UK, and Colombia, all of whom are descended from West African or Southwest African groups, are more genetically related to European descendants in these countries than either would be to Khoisan or Rainforest Hunter-Gatherers. This is despite the fact that Khoisan, Rainforest Hunter-Gathers, and West African individuals would all likely be considered as Black in these countries, based on their phenotype and geographical origins, whereas European descendants would be considered as White.

The seeming contradiction of racially and ethnically stratified pharmacogenomic variation versus distinct global patterns of human genetic variation can be resolved by understanding how race and ethnicity are defined at the local level. As described in the previous section, different countries define race and/or ethnic groups based on the demographic characteristics of their own populations, which are in turn shaped by country-specific patterns of colonization and immigration [4,5,6,7]. Definitions of race and ethnicity are also informed by political considerations, including advocacy by groups that seek official recognition, such as the Hispanic/Latino group in the US, which was only fully recognized as distinct ethnicity starting with the 1980 census [30]. The distinct mix of native and immigrant groups that shape the demography of each country can be considered to represent an uneven sampling of (somewhat) continuous global genetic diversity, and it is this uneven sampling that yields meaningful genetic differences between socially constructed groups. This process can be illuminated using a number line analogy introduced by Joseph Graves [31]. Considering human genetic variation as continuously distributed along a number line from one to ten, if a country’s population is generated by sampling discontinuously, around the values of one, five, and ten for instance, then this sampling would yield genetically distinct groups from an underlying continuous distribution of genetic variation. While this analogy is an oversimplification, it does reflect the kind of historical processes that generated the populations of modern, cosmopolitan countries and thereby explains how locally defined racial and ethnic groups can differ genetically in ways that are distinct from global patterns of human genetic variation.

## 4. Pharmacogenomics and Modifiable Risk Factors

### The Characterization of Pharmacogenomic Variants Provides a Means to Elucidate Modifiable Risk Factors in Support of Health Equity

Clinical applications of pharmacogenomics are closely tied to the concept of risk stratification. A risk factor is anything that increases the chance of developing a health condition or disease. The characterization and measurement of risk factors can be used to inform disease prevention strategies and treatment programs. Risk factors can be characterized as environmental, behavioral, physiological, demographic, or genetic. Environmental risk factors include chemical exposures and social relationships, behavioral risk factors include smoking, diet, and physical activity, and physiological risk factors include weight, blood pressure, and cholesterol. Demographic risk factors relate primarily to race, ethnicity, and sex, and genetic risk factors are based on individuals’ genetic makeup, i.e., their collection of genetic variants.

Effective health interventions require modifiable risk factors. Environmental, behavioral, and physiological risk factors are all (potentially) modifiable, and it makes perfect sense for public health strategies and medical interventions to focus on modifiable risk factors of this kind. A corollary, and seemingly reasonable, critique of demographic and genetic risk factors is that they are not (readily) modifiable. Demographic characteristics are considered to be socially ascribed, meaning that they are assigned at birth or involuntarily later in life. Since socially ascribed characteristics are neither chosen nor earned, they are typically not considered to be modifiable. Genetic variants are inherited at birth and through the process of development manifest in the ~37 trillion cells of the human body. Thus, genetic risk factors are also non-modifiable. It should be noted that genetic risk factors may become partially modifiable in the not-too-distant future owing to developments in CRISPR or other gene editing approaches, but for the moment these technologies are not widely available in the healthcare setting. The key point with respect to epidemiology and health is that a focus on the discovery of non-modifiable demographic or genetic risk factors may not directly inform disease prevention and treatment.

Pharmacogenomics provides a direct link between non-modifiable demographic or genetic risk factors and modifiable environmental risk factors. This is because pharmacogenomic variants are defined as a special case of gene-by-environment interactions, specifically the interaction between genetic variants and medications used to treat disease. And medications are of course a kind of chemical environmental exposure. The effects of medications—with respect to dosage, efficacy, and toxicity—vary based on the presence or absence of specific pharmacogenomic variants. In this sense, the presence of pharmacogenomic variants in individual patients, or their enrichment in certain demographic groups, can suggest environmental modifications, i.e., changing or modifying drug prescriptions, that lead to improved health outcomes. The way that pharmacogenomic variants are assayed—directly in the case of pharmacogenetic tests or indirectly in the case of variant population frequencies—will determine the level of intervention. Precision medicine approaches require pharmacogenomic information for individual patients, whereas precision public health relies on pharmacogenomic information gleaned at the population level. While demographic data, for race and ethnicity in particular, are not reliable proxies for individual-level pharmacogenetic variation, they can serve as valuable guides for population-level screening and as risk factors for patient group stratification. This crucial distinction is discussed at greater length in the following section on precision medicine versus precision public health.

## 5. Precision Medicine Versus Precision Public Health

### Population Pharmacogenomics Is an Essential Component of Precision Public Health, Where the Focus Is on Population-Level Variation as Opposed to Individual-Level Differences

Precision (genomic) medicine is an emergent medical discipline that involves the use of patients’ genomic information to inform their clinical care, with respect to disease diagnosis, prognosis, and treatment [32,33]. Pharmacogenomic data is foundational to the treatment arm of precision medicine, which aims to deliver “the right medicine, to the right patient, at the right time”. In other words, precision medicine is squarely focused on individual-level pharmacogenomic information, i.e., the presence or absence of specific drug-associated variants in any given patient. Precision public health, on the other hand, relies on population-level data to inform public health strategies [34,35,36]. As it relates to pharmacogenomics, precision public health aims for “the right intervention, to the right population, at the right time.” Population pharmacogenomic profiles—data on the relative frequencies of pharmacogenomic variants within and between populations of interest—can be used to guide precision public health initiatives. For example, as we demonstrated in our Colombian study, population pharmacogenomic profiles can be used to target the provisioning of resources and testing where they are more likely to lead to improved health outcomes [21].

The distinction between precision medicine versus precision public health is key to understanding the utility, or lack thereof, of race and ethnicity as proxies for pharmacogenomic variation. Race and ethnicity are poor proxies for individual-level genetic variation and thus should not be used in place of direct pharmacogenomic testing for precision medicine. However, race and ethnicity do effectively stratify pharmacogenomic variation at the population level and therefore can be used to inform precision public health. This distinction is exemplified by the story of the nationwide rollout of the HIV medication efavirenz in Zimbabwe [37].

In 2015, Zimbabwe followed the World Health Organization’s recommendation for HIV public health programs and switched from a three-drug cocktail to the single combination pill efavirenz. This change led to widespread adverse effects, including hallucinations, depression, and suicidal tendencies, with thousands of patients quickly abandoning treatment. It turns out that these adverse effects were associated with a pharmacogenomic variant that is found at anomalously high frequency in the Zimbabwe population. Approximately 20% of the population has two copies of the recessive efavirenz adverse effect associated variant, a fact which had been reported by Collen Masimirembwa, a local scientist working in the capital city Harare seven years earlier [38]. If Masimirembwa’s advice on efavirenz dosing tailored to the local population had been heeded, the country could have avoided a public health crisis.

This example underscores the distinction between the individual-level prediction accuracy of population labels compared to their utility for population-level stratification. If Zimbabwe is taken as a population label and used as a surrogate to predict individual patients’ adverse reactions to efavirenz, based on population pharmacogenomic data, it would have an extremely low prediction accuracy of 20% in the country. However, if the same pharmacogenomic data were used to inform public health interventions, they could prove to be extremely useful. Precision public health can leverage population pharmacogenomic data of this kind to inform population-wide treatment programs, directing pharmacogenomic testing where it is most needed and avoiding drugs that are predicted to cause adverse effects in a large fraction of the population. Race and ethnicity labels provide sufficient pharmacogenomic risk stratification to support the precision public health approach. We explore our own results on the relationship between race, ethnicity, and adverse drug reactions in the following section.

## 6. Adverse Drug Reactions

### The Mitigation of Adverse Drug Reactions Is an Area Where Population Pharmacogenomics Can Have a Direct and Immediate Impact on Public Health

Toxicity-associated pharmacogenomic variants are linked to adverse reactions to numerous widely prescribed medications. As we have shown previously, differences in the frequencies of toxicity-associated pharmacogenomic variants between racial and ethnic groups have important implications for public health. Our approach to studying the implications of race and ethnicity for adverse drug reactions relies on the excess number of predicted adverse reactions per one thousand patients [24]. Given the population frequency of toxicity-associated variants, along with their documented effect mode as either dominant requiring a single copy of the toxicity-associated allele or recessive requiring two copies of the toxicity-associated allele, the number of predicted adverse effects can be calculated. Population stratification, using race, ethnicity, or any other population grouping approach, can then be used to calculate group differences in the number of predicted adverse reactions for medications with toxicity-associated variants.

Application of this approach to US racial and ethnic groups, using the *All of Us* and *HRS* cohorts, yields striking results [24,25]. The Black group shows up to 726 predicted excess adverse drug reactions per 1000 patients compared to the majority White group for the dominant effect mode and up to 635 excess adverse reactions for the recessive effect mode. The Hispanic group, which is more ancestrally similar to the majority White group, shows up to 351 predicted adverse drug reactions per 1000 patients compared to the White group for the dominant effect mode and up to 286 excess adverse reactions for the recessive effect mode. 

By way of example, a pharmacogenomic variant in the *CYP3A4* (Cytochrome P450 Family 3 Subfamily A Member 4) gene has been associated with adverse reactions to methadone, which is used to treat heroin-dependent patients. In the *All of Us* cohort, the toxicity-associated allele of this variant (dbSNP rs4646437) shows high frequency in the Black group (0.73) compared to the majority White group (0.11), and it exerts a dominant effect on adverse reactions to methadone. Given these allele frequency differences between groups, the dominant effect mode predicts 719 excess adverse drug reactions per 1000 patients in the Black group compared to the majority White group.

A pharmacogenomic variant in the *VKORC1* (Vitamin K Epoxide Reductase Complex Subunit 1) gene has been associated with anticoagulation and excess bleeding in patients treated with the blood thinners warfarin and phenprocoumon. In the *All of Us* cohort, the toxicity-associated allele of this variant (dbSNP rs9923231) shows high frequency in the Asian (0.67) group compared to the majority White group (0.34), and it exerts a dominant effect on adverse reactions to warfarin and phenprocoumon. This allele frequency difference, given the dominant effect mode, predicts 327 excess adverse drug reactions per 1000 patients in the Asian group compared to the majority White group.

A pharmacogenomic variant in the *DPYD* (Dihydropyrimidine Dehydrogenase) gene has been associated with toxic side effects to the chemotherapeutic agents fluorouracil and capecitabine. In the *All of Us* cohort, the toxicity-associated allele of this variant (dbSNP rs1801265) shows high frequency in the Black (0.40) group compared to the majority White group (0.22), and it has been report to exert both recessive and dominant effects on toxicity. This allele frequency difference predicts 112 excess adverse drug reactions per 1000 patients in the Black group compared to the majority White group under the recessive effect mode and 248 excess adverse reactions under the dominant mode.

A pharmacogenomic variant in the *ERCC1* (Excision Repair 1, Endonuclease Non-Catalytic Subunit) gene has been associated with adverse reactions to nine different drugs, including cisplatin and oxaliplatin chemotherapy drugs. In the *HRS* cohort, the toxicity-associated allele of this variant (dbSNP rs11615) shows high frequency in the Black (0.88) and Hispanic (0.64) groups compared to the majority White group (0.37). These large allele frequency differences, given the recessive effect mode, yield predictions of up to 638 excess adverse drug reactions per 1000 patients in the Black group and up to 273 excess adverse reactions in the Hispanic group compared to the majority White group.

These findings on race, ethnicity, and adverse drug reactions also have implications for the clinical relevance of human genetic variance components. In 1972, Richard Lewontin found that the majority of human genetic variation was found within (85%) rather than between (15%) racial groups [39]. This fundamental result has been replicated many times since, and it is often taken to support the irrelevance of racial classification to human genetic variation [40,41]. This is consistent with the notion that race and ethnicity are purely social constructs with little or no biological significance [42]. Here, it must be reiterated that observed patterns of pharmacogenomic variation, particularly as they relate to adverse drug reactions, clearly support the clinical relevance of race and ethnicity [21,22,24,25]. In fact, we previously showed that when pharmacogenomic variation is partitioned exactly in the way that Lewontin and others have described, 85% within-group variation and 15% between-group variation, there can be up to 700 excess adverse drug reactions per 1000 patients predicted for the recessive effect mode and as many as 300 adverse reactions predicted for the dominant mode [24]. In other words, a high relative excess of within- versus between-group genetic variation, as is almost always seen for human populations, does not preclude the utility of race and ethnicity for pharmacogenomic risk stratification.

## 7. Conclusions: Embrace Genomic Diversity for Health Equity

### A Reckoning with the Implications of Population Pharmacogenomic Diversity Is a Prerequisite for Health Equity

To conclude this perspective, we would like to emphasize once again the relevance of human genomic diversity for research efforts and clinical applications aimed at promoting health equity [43,44]. The extent of actionable pharmacogenomic differences between racial and ethnic groups flies in the face of an emerging orthodoxy, which maintains that race and ethnicity are social constructs with no biological or clinical relevance [3,42]. The ascendance of this strict social constructionist view of race and ethnicity points to a fundamental contradiction at the heart of genomics research efforts. On the one hand, there is an increasing awareness of the need to diversify genomics research cohorts in support of health equity, driven by the realization that the current bias towards European ancestry cohorts will limit the global reach and impact of genomic medicine [45,46,47,48]. This view is exemplified by the National Human Genome Research Institute’s (NHGRI) pangenome initiative, which aims to expand the human reference genome to include representation from scores, and ultimately hundreds, of genome sequences from diverse human populations [49]. On the other hand, more and more genomics researchers are simultaneously advocating for the elimination of race and ethnicity in genomics research and for developing methods that emphasize the genetic sameness among populations rather than their differences [50,51,52,53,54]. This worldview stresses a continuum of human global diversity with population groups representing operationally defined abstractions rather than any real underlying structure in the data. There are even efforts to remove widely used terms like ancestry and admixture from the scientific lexicon, so as to avoid essentializing human genetic variation [55,56]. This body of work is part of a larger trend, which Jerry Coyne and Luana Maroja recently described as “the ideological subversion of biology” [57]. Much as we have done here, Coyne and Maroja soundly reject the ideologically motivated claim that there is no empirical value in studying genetic differences between races, ethnic groups, or populations, and they point out the threat that ideological research prerogatives of this kind pose to open scientific inquiry.

We contend that scientists should be able to accommodate more than one view on the constitution of racial and ethnic groups. Clearly, these groups are socially constructed, given that their boundaries and composition are delineated by humans. Accordingly, they differ with respect to non-genetic factors that are relevant to health outcomes, such as diet, lifestyle, and socioenvironmental factors. And just as clearly, socially defined racial and ethnic groups can and do differ genetically in ways that are relevant to disease treatment decisions. It follows that efforts to eliminate race and ethnicity from genomics research and clinical considerations, however well-intentioned, will ultimately do more harm than good, exacerbating rather than ameliorating existing health disparities. A more nuanced approach to race and ethnicity, one which recognizes both the social and genetic dimensions of these groups, can be effectively leveraged to support health equity following the roadmap laid out by NIMHD, promoting fairness and opportunity in disease treatments and health outcomes.

## Figures and Tables

**Figure 1 genes-14-01840-f001:**
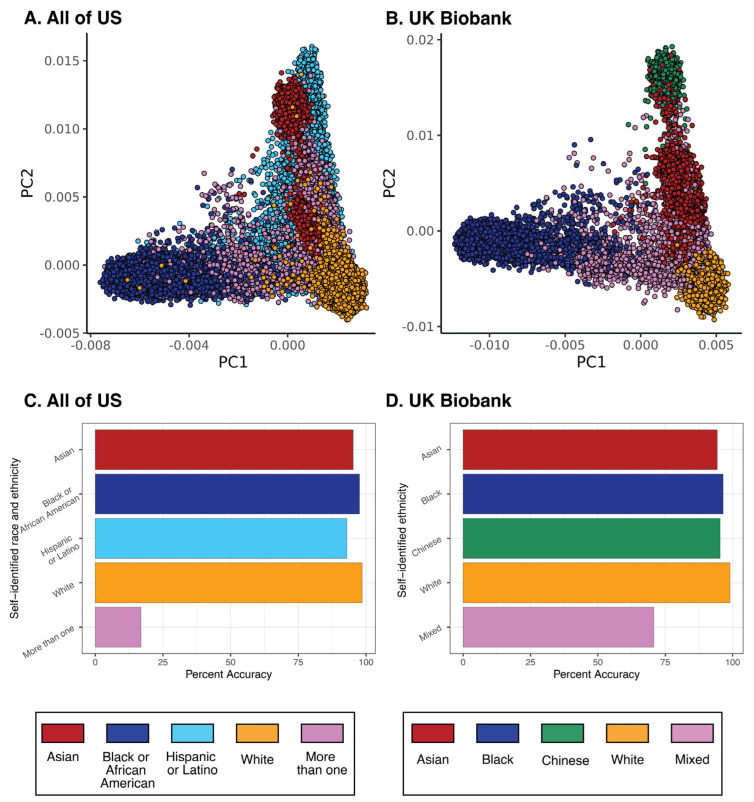
Pharmacogenomic variation corresponds to self-identified race and ethnicity. Principal component analysis for (**A**) *All of Us* and (**B**) *UKB* based on pharmacogenomic variants. Individual participants are shown as circles and color-coded by their self-identified race or ethnicity (see key below plot). The percent accuracy with which pharmacogenomic variation predicts race and ethnicity is shown for (**C**) *All of Us* and (**D**) *UKB*.

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
