# Peer review of "Population Pharmacogenomics for Health Equity"

_genes, 2023, doi:10.3390/genes14101840_

Round 1

Reviewer 1 Report

This is an interesting manuscript elaborating on the relevance of population pharmacogenomics for health equity. The authors mention the differences observed between the social and genetic implications in the definition of racial or ethnic groups, but they highlight the relevance of these differences in the clinical outcome to pharmacological treatment. I have only minor comments:

- Sometimes the language of the manuscript lacks scientific rigor. I think it is important to maintain scientific soundness and to add particular references to enrich the manuscript. Mainly for the ideas described in lines 116-119 and 221-227.

- I consider that the pharmacogenomic variants described by the authors in section 6 are not the best examples to prove their hypothesis. I agree that the frequencies' differences are relevant between populations, but the clinical implications of both pharmacogenomic biomarkers are really questionable. 

- I agree with the point of view of the authors. There are important population differences in the frequencies of relevant pharmacogenetic biomarkers that make it necessary to consider them in public health decisions. However, studies investigating the clinical implications of those variants on the safety and efficacy of the drugs including the mentioned populations are still necessary. Clinical studies including those minority groups should be mentioned and discussed, since it is necessary to show that the consideration of pharmacogenomic biomarkers for the underrepresented groups will really make a difference in their health improvement, not only supported by numbers (frequencies). In addition, it is important to mention that several of these groups are treated with traditional medicine, or have particular food intake habits, or have different health needs than other populations. All of these demographic or non-genetic factors should not be neglected.

Author Response

Reviewer #1

Comment #1: This is an interesting manuscript elaborating on the relevance of population pharmacogenomics for health equity. The authors mention the differences observed between the social and genetic implications in the definition of racial or ethnic groups, but they highlight the relevance of these differences in the clinical outcome to pharmacological treatment.

Response #1: We appreciate the reviewer’s positive response to our manuscript.

I have only minor comments:

Comment #2: Sometimes the language of the manuscript lacks scientific rigor. I think it is important to maintain scientific soundness and to add particular references to enrich the manuscript. Mainly for the ideas described in lines 116-119 and 221-227.

Response #2: As suggested, we have reviewed the manuscript to ensure that the language is scientifically rigorous throughout.  Also, we have included references on the pulse oximeter as suggested (see references 21-23).

Comment #3: I consider that the pharmacogenomic variants described by the authors in section 6 are not the best examples to prove their hypothesis. I agree that the frequencies' differences are relevant between populations, but the clinical implications of both pharmacogenomic biomarkers are really questionable.

Response #3: As suggested, we have included additional examples in section six of biomarkers with better known clinical implications (see lines 346-361).  These include two distinct toxicity-associated variants in the widely studied pharmagene DPYD (Dihydropyrimidine Dehydrogenase), both of which correspond to the highest level of PharmGKB evidence 1A.

Comment #4: I agree with the point of view of the authors. There are important population differences in the frequencies of relevant pharmacogenetic biomarkers that make it necessary to consider them in public health decisions. However, studies investigating the clinical implications of those variants on the safety and efficacy of the drugs including the mentioned populations are still necessary. Clinical studies including those minority groups should be mentioned and discussed, since it is necessary to show that the consideration of pharmacogenomic biomarkers for the underrepresented groups will really make a difference in their health improvement, not only supported by numbers (frequencies). In addition, it is important to mention that several of these groups are treated with traditional medicine, or have particular food intake habits, or have different health needs than other populations. All of these demographic or non-genetic factors should not be neglected.

Response #4: The reviewer raises a very important point that we had not considered.  As suggested, we included a statement that racial and ethnic groups “differ with respect to non-genetic factors that are relevant to health outcomes, such as diet, lifestyle, and socioenvironmental factors” to the revised Conclusion section (see lines 419-421).

Reviewer 2 Report

The article offers an interesting perspective on how precision medicine intersects with public health.  The variation of genetics within ethnic groups vs between ethnic groups and the role related to pharmacogenetics is interesting but I find the work lacks a clear direction to lead the readers to its conclusion that "population pharmacogenomic diversity is a prerequisite for health equity".

The work can be shortened and made more clear.  Some of the examples with different technology are interesting and show the biases in research and marketing health technology, including impacts on patient outcomes - but they introduce some noise into the overall message of the paper.

The concepts of ethnicity as a social construct can be made shorter as bouncing back to the topic introduces some noise and takes away from the messaging in the paper.

The written English is fine - the overall structure of the paper can be shortened, which would make the message and conclusion more clear.

Author Response

Reviewer #2

Comment #1: The article offers an interesting perspective on how precision medicine intersects with public health.  The variation of genetics within ethnic groups vs between ethnic groups and the role related to pharmacogenetics is interesting but I find the work lacks a clear direction to lead the readers to its conclusion that "population pharmacogenomic diversity is a prerequisite for health equity".

Response #1: We appreciate the reviewer’s positive response to our manuscript, and we have endeavored to make the connection to "population pharmacogenomic diversity is a prerequisite for health equity".

Comment #2: The work can be shortened and made more clear.  Some of the examples with different technology are interesting and show the biases in research and marketing health technology, including impacts on patient outcomes - but they introduce some noise into the overall message of the paper.

The concepts of ethnicity as a social construct can be made shorter as bouncing back to the topic introduces some noise and takes away from the messaging in the paper.

The written English is fine - the overall structure of the paper can be shortened, which would make the message and conclusion more clear.

Response #2: We appreciate the reviewer’s suggestions to shorten and focus the work, particularly as it relates to the concept of ethnicity as a social construct.  We have endeavored to focus the presentation of our perspective as suggested.  Nevertheless, we feel that it may be of substantial use to readers to include this discussion, particularly in light of the emerging orthodoxy that race and ethnicity are social constructs with no biological or scientific meaning.  For example, the Journal of the American Medical Association recently published updated guidance on the reporting of race and ethnicity, where they state that “Race and ethnicity are social constructs, without scientific or biological meaning” (JAMA 2021; 326:621).  This statement is empirically false, particularly as it relates to pharmacogenomic variation.  This orthodoxy has caused much confusion, particularly in light of the fact that clinically relevant genetic differences between racial and ethnic groups do exist, and it serves as a potential barrier to more equitable health care that considers patients ancestral backgrounds as they relate to pharmacogenomics.  For these reasons, we feel that it is worthwhile to elaborate on how this confusion has found its way to the scientific literature and to paint a more nuanced and accurate picture of the relationship between race, ethnicity, and pharmacogenomic variation.  (See also response to reviewer #3, comment #1).

Reviewer 3 Report

Recently, there are studies that suggest that it is necessary to consider health equity due to genetic differences without individual choice. I fully agree with this view. This study is in the same vein, but instead of a gene, a more detailed term called pharmacogenomics is used. However, the overall conclusion is that genetic diversity should be considered for health equity. Furthermore, there are precedent studies that actively raised what is needed to improve equity.

Therefore, it is necessary to clarify what is the difference between this paper and recent studies like this one. To this end, the following comments are presented.

1. It would be better to read if the previous studies that raised the issue of health equity due to genetic differences were summarized in a table and the related genes or pharmacogenetic factors were written.

2. It is also recommended to briefly present the author's opinion by actively using photographs or images.

In the past, genetic research has made many advances. As much remains to be developed in the future, further genetic differences must be identified. Therefore, the opinions and arguments of this study are valuable as continuous efforts are needed to reduce the health gap.

Author Response

Reviewer #3

Comment #1: Recently, there are studies that suggest that it is necessary to consider health equity due to genetic differences without individual choice. I fully agree with this view. This study is in the same vein, but instead of a gene, a more detailed term called pharmacogenomics is used. However, the overall conclusion is that genetic diversity should be considered for health equity. Furthermore, there are precedent studies that actively raised what is needed to improve equity.

Therefore, it is necessary to clarify what is the difference between this paper and recent studies like this one. To this end, the following comments are presented.

It would be better to read if the previous studies that raised the issue of health equity due to genetic differences were summarized in a table and the related genes or pharmacogenetic factors were written.

Response #1: We appreciate the reviewer’s comment regarding the need to survey previous work.  As this article is an invited perspective article, we have chosen to focus on our own perspective and contributions rather than conduct a comprehensive review of previous work.  Nevertheless, we have addressed the reviewer’s comments and revised our manuscript as described below.

We would also like to point out that what makes our paper different from recent surveys on the topic is our focus on the distinction between the socially defined concepts of race and ethnicity and the biological reality of pharmacogenomic differences between socially defined groups.  This discussion takes on some of the misconceptions around this topic that have been propagated in the recent scientific literature and have the potential to serve as a barrier to health equity.  (See also response to reviewer #2, comment #2).

Comment #2: It is also recommended to briefly present the author's opinion by actively using photographs or images.

Response #2: As suggested, we include an image that highlights differences in the frequency of pharmacogenomic variants associated with adverse drug reactions between race and ethnic groups based on our previous studies on the All of Us Research Project and UK Biobank in the revised manuscript (see Figure 1 on page 4).

Comment #3: In the past, genetic research has made many advances. As much remains to be developed in the future, further genetic differences must be identified. Therefore, the opinions and arguments of this study are valuable as continuous efforts are needed to reduce the health gap.

Response #3: We appreciate the reviewer’s positive comments on the manuscript.

Reviewer 4 Report

The present article, entitled "Population Pharmacogenomics for Health Equity," is a perspective that shows the gains related to precision public health in comparison with precision medicine. The authors also contend that population pharmacogenomics studies need to be leveraged, with a focus on health equity as well as the elaboration of public health policies, to apply the correct intervention to the right population at the right time. Moreover, two sociodemographic variables (race and ethnicity) were highlighted by the authors as useful for population pharmacogenomics studies and, consequently, for precision public health.

The article covered an intriguing subject and was well-written, concise, and objective. Great job!

Minor issues: in some parts of the manuscript, the authors used the expressions "effect mode", "dominant effect mode", "recessive effect mode", "recessive effect model," or "dominant model" (lines 322, 331, 332, 334, 335, 342, 364, and 365). Does the correct word be mode or model? Please check and correct it where appropriate.

Author Response

Reviewer #4

Comment #1: The present article, entitled "Population Pharmacogenomics for Health Equity," is a perspective that shows the gains related to precision public health in comparison with precision medicine. The authors also contend that population pharmacogenomics studies need to be leveraged, with a focus on health equity as well as the elaboration of public health policies, to apply the correct intervention to the right population at the right time. Moreover, two sociodemographic variables (race and ethnicity) were highlighted by the authors as useful for population pharmacogenomics studies and, consequently, for precision public health.

The article covered an intriguing subject and was well-written, concise, and objective. Great job!

Response #1: We thank the reviewer for their positive comments regarding our manuscript.

Comment #2: Minor issues: in some parts of the manuscript, the authors used the expressions "effect mode", "dominant effect mode", "recessive effect mode", "recessive effect model," or "dominant model" (lines 322, 331, 332, 334, 335, 342, 364, and 365). Does the correct word be mode or model? Please check and correct it where appropriate.

Response #2:  Mode is the correct term here, and we have replaced model with mode as suggested by the reviewer (see section 6. Adverse drug reactions).

Round 2

Reviewer 1 Report

Thank you for the modifications accordingly the comments. 

Author Response

We are glad the reviewer is satisfied with our responses and modifications to the manuscript.

Reviewer 2 Report

Thank for the modifications in the manuscript and your comments.  They have helped with the interpretation of the work.

The caption / legend for Figure 1 should be expanded to explain what is being depicted in the figure.

Author Response

We are glad that the reviewer is satisfied with our responses and modifications to the manuscript.

As suggested, we have provided a legend for Figure 1 to explain what is being depicted in the figure.